# A Kolmogorov Complexity Approach to Generalization in Deep Learning

## Abstract

Deep artificial neural networks can achieve an extremely small difference between training and test accuracies on identically distributed training and test sets, which is a standard measure of generalization. However, the training and test sets may not be sufficiently representative of the empirical sample set, which consists of real-world input samples. When samples are drawn from an underrepresented or unrepresented subset during inference, the gap between the training and inference accuracies can be significant. To address this problem, we first reformulate a learning algorithm as a procedure for searching for a source code that maps input features to classes. We then derive a necessary and sufficient condition for generalization using a universal cognitive similarity metric, namely information distance, based on Kolmogorov complexity. Using this condition, we formulate an optimization problem to learn a more general classification function. To achieve this end, we extend the input features by concatenating encodings of them, and then train the classifier on the extended features. As an illustration of this idea, we focus on image classification, where we use channel codes on the input features as a systematic way to improve the degree to which the training and test sets are representative of the empirical sample set. To showcase our theoretical findings, considering that corrupted or perturbed input features belong to the empirical sample set, but typically not to the training and test sets, we demonstrate through extensive systematic experiments that, as a result of learning a more general classification function, a model trained on encoded input features is significantly more robust to common corruptions, e.g., Gaussian and shot noise, as well as adversarial perturbations, e.g., those found via projected gradient descent, than the model trained on uncoded input features.

## 1 Introduction

Generalization error in deep learning is typically defined as the difference between training and test errors measured on identically distributed training and test sets. This traditional approach fails to take into account how representative these sets are of the empirical sample set from which input samples are drawn at inference time. When the training and test sets are not sufficiently representative of the empirical sample set, the difference between training and inference errors can be significant, thus rendering the learned classification function ineffective. The lack of the latter kind of generalization results in unreliable decisions, raising questions about how robust, fair, and safe a learned classification function is (Varshney & Alemzadeh, 2017).

A natural question then arises: is there a necessary and sufficient condition ensuring that deep learning classifiers generalize in this broader sense? If so, how can this condition be satisfied in a real-world setting? To answer these questions, we draw on algorithmic information theory, which proposes a complexity measure, Kolmogorov complexity, as the absolute information content of any object, e.g., a computer program, function, or set. After deriving a necessary and sufficient condition for generalization using the *information distance* (Bennett et al., 1998), which is a universal cognitive similarity metric based on Kolmogorov complexity, and formulating an optimization problem for generalization, we turn our attention to coding theory in order to learn a more general classification function by extending the input features to a classifier with systematically generated encodings of the original features.

## 1.1 OUR CONTRIBUTIONS

For a classification task, we assume that there exists a true classification function. Given training and test sets, neither of which are sufficiently representative of the the empirical sample set from which input samples are drawn during inference, a learning algorithm is asked to find the true classification function. In this work, we study how well the learned classification function generalizes with respect to the true classification function. In other words, we study the problem of how to minimize the generalization error, which we define as the difference between the training error and inference error measured on the empirical sample set, as opposed to the difference between the training error and test error. We use robustness to both common corruptions and adversarial robustness to measure how well a learned classification function generalizes on the empirical sample set, which contains corrupted or perturbed samples.

**Universal cognitive similarity metric.**   In order to find a necessary and sufficient condition for generalization in deep learning, we use the normalized information distance. A key finding in algorithmic information theory is that the normalized information distance is a universal cognitive similarity metric: the normalized information distance between two objects minorizes any other admissible distance up to an additive logarithmic term (Bennett et al., 1998). In other words, although different learning algorithms will pick up on different dominating input features, depending on the classification task that they perform, every such dominating feature will be detected by the normalized information distance.

**Classification function as a source code.**   We formulate a learning algorithm as a procedure for searching for a source code based on training examples. We show that the learned classification function is a lossy compressor: the classifier discards some information. The input features thus cannot be recovered from the class label. We use the normalized information distance between the true source code (true classification function) and the learned source code (learned classification function) to find a necessary and sufficient condition ensuring generalization, and then formulate the problem of learning a more general classification function as an optimization problem.

**Compression-based similarity metric.**   The normalized information distance provides the theoretical tools needed to learn more general source codes, but in practice the normalized information distance is not effectively computable. We therefore use a compression-based similarity metric (Cilibrasi & Vitányi, 2005) based on a real-world compressor to approximate this theoretical construct. Specifically, we use the normalized compression distance between the true source code and learned source code to derive an effectively computable condition on the compressed size of the learned source code to identify encodings of the input features that help to learn a more general source code.

**Encoding input features.**   In a typical communication system, a source code is followed by a channel code which is then followed by a physical channel. In this paper, the learned source code (learned classification function) is preceded by one or more input codes that help ensure the learned classifier is more general by generating relations between input features that are not captured by the set of available input features. In order to showcase our findings for a specific classification task, we use channel codes on the input features for CIFAR-10 image classification. Precisely, we use a four-dimensional (4-D) five-level pulse-amplitude modulation (5-PAM) trellis-coded modulation (TCM) scheme (Ungerboeck, 1982; Hatamian et al., 1998; IEEE, 2015) to systematically generate multiple encodings of the set of available input features. In doing so, we enable the deep neural network (DNN) to learn information from the empirical sample set which it could not learn from the uncoded input features alone. The generalization error is thereby reduced.

**The impact of generalization.**   Through image classification experiments, we show that a model trained on arbitrarily encoded input features is significantly more robust to common corruptions, such as Gaussian noise and shot noise, and to adversarial perturbations, like those generated via projected gradient descent (PGD) (Madry et al., 2018), than a model trained on uncoded input features.

**The role of code design.**   The code used on the input features can be designed in various ways for a classification task, and designing input codes is an important step to learning a more general classification function from the set of available input features. We show that merely increasing the

number of input channels of a DNN does not confer any robustness to Gaussian noise or to PGD. How to design efficient input codes to build encoded DNNs is an intriguing research direction for achieving generalization in deep learning.

## 1.2 RELATED WORK

The literature on generalization, e.g. (Zhang et al., 2017; Neyshabur et al., 2017), is largely concerned with minimizing the generalization error, defined as the difference between training and test errors measured on identically distributed training and test sets. Minimizing this form of generalization error does not address the problem of generalizing to input samples drawn from an empirical sample set of which the training and test sets are not sufficiently representative, as we do herein.

In this subsection, we compare our work with domain-generalization, domain-adaptation, and data-augmentation techniques to highlight their differences. There is a substantial body of literature on domain generalization (Muandet et al., 2013; Li et al., 2017; Motiian et al., 2017; Shankar et al., 2018), which aims to better generalize to unknown domains by training on samples drawn from different domains, not a single source, which is a limitation that our work does not have. In this work, there is no need to draw training samples from a different domain. We show that encoding the given training set enables a DNN to learn different relations between features that it could not learn from the uncoded training set alone.

There has also been much work on domain adaptation (Daumé III & Marcu, 2006; Saenko et al., 2010; Ganin & Lempitsky, 2015; Tzeng et al., 2017; Sun & Saenko, 2016; Morerio et al., 2018; Volpi et al., 2018a) that addresses the problem of generalization to *a priori* fixed target domains, which is a different approach from ours because these algorithms need to access samples from the target distributions during an adaptation phase. Importantly, our approach does not require accessing new samples during an adaptation phase in order to achieve generalization to the empirical sample set. Similar to the domain adaptation work, there has been some work on adversarial training (Goodfellow et al., 2015; Lee & Raginsky, 2018; Sinha et al., 2018), which aims to achieve robustness (Zhang et al., 2019) to adversarial perturbations by using training samples perturbed by a specific adversarial-perturbation method. Adversarial training can be computationally costly because it requires generating adversarially perturbed training samples in each epoch of training, unlike in our work where input encodings need to be generated only once before training. Furthermore, as there are numerous adversarial-perturbation methods (Goodfellow et al., 2015; Kurakin et al., 2017b; Madry et al., 2018), an adversarially trained DNN does not necessarily generalize well to samples subjected to an adversarial perturbation method that was not used for training (Madry et al., 2018).

There is also a substantial body of work on data-augmentation techniques (LeCun et al., 1995; Volpi et al., 2018b), which perform simple label-preserving transformations of the training samples to provide a DNN with additional data points to learn from. In this work, we do not generate new samples to increase the diversity of the training set; instead, we take a theoretically-grounded approach to extend the input features with their encodings in order to enable a DNN to learn a sufficiently complex classification function from the set of available input samples.

## 2 ALGORITHMIC INFORMATION-THEORETIC APPROACH TO DEEP LEARNING

Our goal is to minimize the generalization error (defined in Appendix B) for a classification task, defined as the difference between training error and inference error, given a training set and a test set, both of which are not sufficiently representative of the empirical sample set from which input samples are drawn at inference time. To accomplish this goal, we derive a necessary and sufficient condition under which a classifier will generalize well, and, based on that condition, cast the search for a classifier with good generalization (defined in Appendix B) as an optimization problem. Our approach requires that we describe and compute the absolute information content of any object, e.g., a computer program, function, or set, in order to determine which of a pair of learned classification functions contains more information of the true classification function. The appropriate tool here is a concept in algorithmic information theory: Kolmogorov complexity (defined in Appendix B). Defining the amount of information in individual objects in terms of their Kolmogorov complexity has the advantage that it refers to these objects in isolation, not as outcomes of a known random source. In contrast, quantifying the amount of information in individual objects based on their Shannon entropy

requires that these objects be treated as members of a set of objects with an associated probability distribution. This understanding is fundamental to our study because applying Shannon entropy to "an estimate of the quantity of information contained in a novel or in the translation of a novel into another language relative to the original" would not be clear (Kolmogorov, 1983). As a DNN may be employed to learn a classification function from a set of features contained in such an object as, for example, a document, image, video, or sound, we study the Kolmogorov complexity of the set of input features, model, and outputs of the DNN.

## 2.1 Normalized Information Distance as Universal Cognitive Similarity

In our quest to find a condition ensuring our running definition of generalization, we require a distance function that measures how similar two objects are in any aspect so we can decide which of two learned classification functions is closer to the true classification function. The closer a learned classification function is to the true classification function, the better its generalization error. This distance function should satisfy the metric (in)equalities in order for it to have a meaning in the context of generalization. For example, this distance function would have to be symmetric; i.e., the distance from object $a$ to object $b$ must be equal to that from object $b$ to object $a$.

The normalized information distance (Bennett et al., 1998) between objects $a$ and $b$, defined as

$$D_{\mathrm{I}}(a, b) = \frac{\max(K(a|b), K(b|a))}{\max(K(a), K(b))} \tag{1}$$

where $K(a)$ denotes the Kolmogorov complexity of object $a$ and $K(a|b)$ denotes the Kolmogorov complexity of object $a$ given $b$, satisfies the metric (in)equalities and is also a universal cognitive similarity metric because $D_{\mathrm{I}}(a, b)$ minorizes all other normalized admissible distances up to a negligible additive error term. This means that all effective similarities between a pair of objects are discovered by the normalized information distance; i.e., two objects that are close according to some effective similarity are also close according to the normalized information distance. The main intuition behind normalizing the information distance $\max(K(a|b), K(b|a))$ is that two larger objects that differ by a small amount are closer than two smaller objects that are different by the same amount: the absolute difference between two objects does not measure similarity as such, but the relative difference does (Cilibrasi & Vitányi, 2005).

## 2.2 Deep-Learning Classifier as a Source Code

A successful DNN distills information useful for its classification task $T$ from its input features $\vec{x}$. In doing so, the DNN has to learn a classification function $f(.)$ from the set $\mathbb{X}^n$ of its input features to an $m$-ary alphabet $\mathbb{A}$ of classes $u$ in such a way that some information in its input features is given less weight in determining its relevance to the class decision $\hat{u}$, and then entirely discarded by the $\arg\max$ operation (Goldfeld et al., 2019). A deep learning classifier is thus acting as a source code $C$ (defined in Appendix B). Proofs of the following mathematical statements are given in Appendix A.

**Lemma 1.** For a classification task $T$ wherein each $n$-dimensional input sample $\vec{x}$ is mapped to a class $u$ drawn from an $m$-ary signal alphabet $\mathbb{A}$, the true output function $f(\cdot)$ of a learning algorithm is a source code $C$ for a multivariate random variable $\vec{X}$.

Lemma 1 reformulates a learning algorithm as a procedure for searching for a source code $C$ for a multivariate random variable $\vec{X}$, which compresses the values that this random variable takes, namely the input samples $\vec{x}$. When a DNN generalizes well with respect to the true classification function $f(\cdot)$, it is able to decide which information in its input features is more relevant to making a particular class decision. A DNN is a lossy compressor when the absolute information content of any of its input samples $\vec{x}$ is larger than that of the class $u$ to which it is mapped.

**Corollary 1.** The true source code $C = f(\cdot)$ of a learning algorithm used for the classification task $T$ is a lossy compressor when the Kolmogorov complexity $K(\vec{x})$ of one of its input samples is larger than the number of bits required to represent the corresponding class $u$.

Corollary 1 formalizes a deep learning classifier as a lossy compressor, so the source code $C$ that corresponds to the true output function $f(\cdot)$ is not uniquely decodable; i.e., its input samples $\vec{x}$ cannot

be recovered from the class $u$ to which they are mapped. A DNN can be trained to learn a source code that generalizes well with respect to the true source code, but first we will analyze the similarity between these two source codes by using the normalized information distance.

Source codes are designed for the most efficient representation of data (Cover & Thomas, 1991). Whether it is designed for a data-transmission or a data-storage system, a source code, whether lossless or lossy, should retain information about the data necessary to accomplish a given task. The same consideration applies to a learning system. The information in the input features of a learning system is represented by the classification function that it learns; thus, a neural network can be viewed as a source code that encodes inputs features for its classification task. The reformulation of a learning algorithm as a procedure for searching for a source code allow us to exploit theoretical results from algorithmic information theory and coding theory for deep learning, thereby avoiding the necessity to reinvent theory that is already established in these fields. Given that source codes are designed for the most efficient representation of data (Cover & Thomas, 1991), we will exploit the duality of a source code and a channel code to learn a classification function that represents the input features more efficiently for the classification task $T$; i.e., a more general classification function. Showing that a deep learning classifier is a non-uniquely decodable source code is also fundamental to understanding that the normalized information distance between the input features and the output cannot be used to derive a condition for generalization in deep learning. This results from the fact that deriving such a condition would require finding the conditional Kolmogorov complexity $K(\vec{x}|y)$ of the input features with respect to the output, which is impossible because the source code is not uniquely decodable; i.e., the program to go from the output to the input features cannot be found. A necessary and sufficient condition for generalization based on the normalized information distance can hence be found only between a learned source code and the true source code.

## 2.3 Achieving Generalization in Deep Learning

The normalized information distance

$$D_{\mathrm{I}}(C, \tilde{C}) = \frac{\max(K(C|\tilde{C}), K(\tilde{C}|C))}{\max(K(C), K(\tilde{C}))}, \tag{2}$$

between the true source code $C$ and learned source code $\tilde{C}$ reveals how general $\tilde{C}$ is with respect to $C$. A necessary and sufficient condition ensuring that learned source code $\tilde{C}_0$ is more general than learned source code $\tilde{C}_1$ with respect to the true source code $C$ is

$$D_{\mathrm{I}}(C, \tilde{C}_0) < D_{\mathrm{I}}(C, \tilde{C}_1), \quad \forall \tilde{C}_0 \neq \tilde{C}_1. \tag{3}$$

Equation 3 is a direct result of using the normalized information distance as a universal cognitive similarity metric to determine whether learned source code $\tilde{C}_0$ or $\tilde{C}_1$ is more general with respect to the true source code $C$. Because the normalized information distance (Bennett et al., 1998) is a metric that uncovers all effective similarities between the true source code and a learned source code, learning a source code that is closer to the true source code $C$ under this metric ensures achieving generalization. The normalized information distance $D_{\mathrm{I}}(C, \tilde{C})$ between the true source code $C$ and the learned source code $\tilde{C}$ must thus be minimized in order to minimize the generalization error.

**Theorem 1.** When a learning algorithm used for the classification task $T$ finds a suboptimal source code $\tilde{C}$ instead of the true source code $C$, the optimization problem for the generalization of $\tilde{C}$ is $\min_{\tilde{C}}(D_{\mathrm{I}}(C, \tilde{C})) = \min_{\tilde{C}} \max(K(C|\tilde{C}), K(\tilde{C}|C))$.

Theorem 1 has formulated the optimization objective for generalization as the minimization of $D_{\mathrm{I}}(C, \tilde{C})$ and states that to achieve generalization we should make the learned function sufficiently complex for the classification task $T$. Theorem 1 states that the Kolmogorov complexity $K(C|\tilde{C})$ of the program that computes how to go from the learned source code $\tilde{C}$ to the true source code $C$ or the Kolmogorov complexity $K(\tilde{C}|C)$ of the program that computes how to go from the true source code $C$ to the learned source code $\tilde{C}$, whichever is larger, must be minimized in order to minimize generalization error. Thus, the goal is to increase the complexity of the learned source code $\tilde{C}$, but not beyond the complexity of the true source code $C$. Therefore, Occam's first razor (Domingos, 1999) still holds: simpler classifiers generalize better than complex ones. However, a classifier that

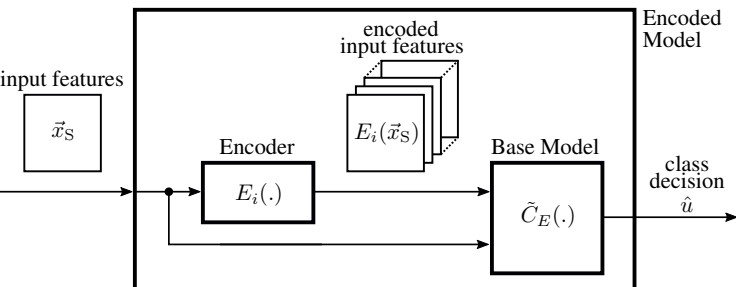

Figure 1: Encoded model architecture. An uncoded model simply feeds the input features to a base model, while the encoded model stacks the input features and encoded features and feeds those to a base model with enough input channels to handle the original and encoded features.

does not perform well on its empirical sample set $\mathbb{X}^n$ is too simple for its classification task. Ideally, the learning algorithm would learn the true source code $C$, achieving the best possible performance metrics determined by its classification task $T$. In practice, because the learning algorithm will see only a small subset $\mathbb{X}_S^n$ of the possible inputs at training time, the learned source code $\tilde{C}$ will be a partial function of the true source code $C$ at perfect training accuracy (that is, when the classifier has sufficient capacity to memorize the training samples (Zhang et al., 2017)). Whether a model is over-fit or under-fit is conventionally determined on a cross-validation set and/or test set that are/is identically distributed with the training set, all of which are subsets of the empirical sample set. Being more general on such a cross-validation set and/or test set does not as such guarantee generalization on the empirical sample set $\mathbb{X}^n$ because the latter may contain corrupted or perturbed samples and/or there may be samples in the empirical sample set that are out of distribution of the cross-validation set and test set. Therefore, whether a model is over-fit or under-fit does not have a consequence for Theorem 1. Next, we target learning a source code that is more general on the empirical sample set $\mathbb{X}^n$, not only on a cross-validation set and/or test set.

In this work, we increase the complexity of the learned source code $\tilde{C}$ by generating $I$ encodings $E_0, E_1, \ldots E_{I-1}$ of the available input features $\vec{x}_S$ that capture relations between the features which are not learned well from the original features, and then append these encodings to the original features. Note that the available input features are denoted by $\vec{x}_S$, which are drawn from the set $\mathbb{X}_S^n$ of available features; i.e., $\vec{x}_S \in \mathbb{X}_S^n$, which is a subset of the empirical sample set $\mathbb{X}^n$. By providing a different view of the relations between the features, the encodings $E_i$ help the learning algorithm to learn a more complex source code $\tilde{C}_E$ whose normalized information distance $D_I(C, \tilde{C}_E)$ to the true source code $C$ is less than $D_I(C, \tilde{C})$. This results in learning a more general source code.

**Theorem 2.** For classification task $T$, a more general suboptimal code $\tilde{C}_E$ is learned from the concatenation $\{\vec{x}_S, E_i(\vec{x}_S)\}$, where $E_i : \mathbb{X}_S^n \to \mathbb{Y}_S^n$ is an encoding of the input sample $\vec{x}_S$ such that $\mathbb{Y}_S^n \not\subseteq \mathbb{X}_S^n$.

The effective capacity of several successful DNN architectures is sufficiently large to memorize the set $\mathbb{X}_S^n$ of available input samples (Zhang et al., 2017). Any encoding $E_i : \mathbb{X}_S^n \to \mathbb{Y}_S^n$, where $\mathbb{Y}_S^n$ is the set of available encoded samples such that $\mathbb{Y}_S^n \not\subseteq \mathbb{X}_S^n$, when concatenated with the uncoded input samples $\vec{x}_S$, thus increases the Kolmogorov complexity of the learned source code, which is now called $\tilde{C}_E$. The task of the source code is to find the most efficient representation of its input data Cover & Thomas (1991). In a typical communication system, the source code compresses the input, then a channel code adds redundancy to guard against noise in the channel, then the encoded information is transmitted over the physical channel. The design goal for the source and channel codes is to achieve the channel capacity (maximum mutual information between the channel input and output). In contrast, Theorem 2 considers a learning system in which an input code is followed by a learned source code, the classification function, and the design goal is for the composition of the input and source codes to generalize as well as possible (see Figure 1). In other words, in a learning system the "physical channel" precedes the source code, and it can be seen as a process whereby the empirical sample set $\mathbb{X}^n$ is reduced to the set $\mathbb{X}_S^n$ of available input samples and/or whereby common corruptions, such as Gaussian noise, and adversarial perturbations, such as those

generated by PGD, are applied to the set of available input samples. Because the "physical channel" comes first in a learning system, there is no access to the set of information bits. Only a subset of these information bits can be accessed, which may have been subjected to common corruptions or adversarial perturbations. It is therefore crucial for a learning algorithm to compress its features while retaining information useful for its classification task. One way to accomplish this is to extend the input features with encodings that capture relations between features that are useful for classification and not captured well by the original set of input features. The classification task $T$ does not change when input features are extended by their encodings because it is defined by the mapping between the input features and the output (Goodfellow et al., 2016), which remains the same because the only input to the encoded model is the uncoded input features (see Figure 1). The encoder is simply a new layer in the encoded model, which is designed from an encoder and an uncoded model.

### 2.4 Approximating Normalized Information Distance by Normalized Compression Distance

The normalized information distance is based on the notion of Kolmogorov complexity, which is not a partial recursive function; i.e., it is not effectively computable. While we can use normalized information distance to analyze whether a source code $\tilde{C}_E$ learned from the concatenation $\{\vec{x}_S, E_i(\vec{x}_S)\}$ of the encoded input samples $E_i(\vec{x}_S)$ with the uncoded input samples $\vec{x}_S$ is more general with respect to the true source code $C$, in practice we may need to approximate the normalized information distance with the normalized compression distance, so we can determine which of any pair of source codes is more general with respect to the true source code $C$.

Based on a real-world compressor, the normalized compression distance (Cilibrasi & Vitányi, 2005)

$$D_{\mathrm{C}}(C, \tilde{C}_E) = \frac{Z(\{C, \tilde{C}_E\}) - \min(Z(C), Z(\tilde{C}_E))}{\max(Z(C), Z(\tilde{C}_E))}, \tag{4}$$

approximates the normalized information distance $D_{\mathrm{I}}(C, \tilde{C}_E)$, where $Z$ is a real-world compressor. Thus, the generalization condition and minimization of $D_{\mathrm{I}}(C, \tilde{C}_E)$ can be cast in effectively computable forms. Note that neither Equation 2 nor Equation 4 is a training criterion, but they specify the normalized information distance and the normalized compression distance between the true source code $C$ and a learned source code, respectively. They are used to derive theoretical results, particularly the use of input codes to achieve generalization as illustrated by experiments in Section 3.

**Proposition 1.** For the classification task $T$, $D_{\mathrm{I}}(C, \tilde{C}_E) < D_{\mathrm{I}}(C, \tilde{C}) \iff Z(\tilde{C}_E) > Z(\tilde{C})$.

Proposition 1 states for classification task $T$ that the compressed size $Z(\tilde{C}_E)$ of the source code $\tilde{C}_E$ learned from the concatenation $\{\vec{x}_S, E_i(\vec{x}_S)\}$ of the encoded input samples $E_i(\vec{x}_S)$ and the uncoded input samples $\vec{x}_S$ is larger than the compressed size $Z(\tilde{C})$ of the source code $\tilde{C}$ learned from the uncoded input samples alone $\vec{x}_S$.

**Proposition 2.** When a learning algorithm used for classification task $T$ finds a suboptimal source code $\tilde{C}_E$ instead of the true source code $C$, the effectively computable optimization problem for the generalization of $\tilde{C}_E$ is $\min_{\tilde{C}_E} D_{\mathrm{C}}(C, \tilde{C}_E) = \max_{\tilde{C}_E} Z(\tilde{C}_E), \forall \tilde{C}_E : Z(\tilde{C}_E) < Z(C)$.

Proposition 2 shows that the compressed size $Z(\tilde{C}_E)$ of the source code $\tilde{C}_E$ learned from the concatenation $\{\vec{x}_S, E_i(\vec{x}_S)\}$ of the encoded input samples $E_i(\vec{x}_S)$ and the uncoded input samples $\vec{x}_S$ must be maximized until it reaches the compressed size $Z(C)$ of the true source code $C$ to learn the most general source code with respect to the true source code $C$ for the classification task $T$. This statement is a consequence of the fact that $\tilde{C}_E$ is a partial function of $C$ at perfect training accuracy. In other words, the source code $\tilde{C}_E$ learned from the concatenation $\{\vec{x}_S, E_i(\vec{x}_S)\}$ of the encoded input samples $E_i(\vec{x}_S)$ and the uncoded input samples $\vec{x}_S$ can be made more general if the encoded input samples $E_i(\vec{x}_S)$ bear information of relations between input features that are not represented by its input samples.

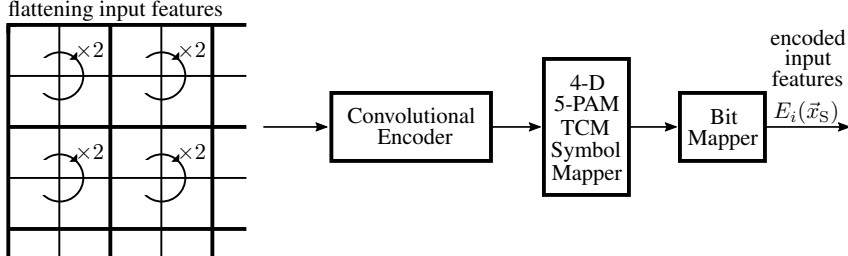

Figure 2: Flattening and encoding input features.

### 2.5 USING CHANNEL CODES ON INPUT FEATURES FOR IMAGE CLASSIFICATION

A channel encoder generates encodings from its input features that enable a classifier to learn relations between these features not captured by the set of available input samples. Concatenated together, these features are then input to a model to produce a class decision. For example, we use a 4-D 5-PAM TCM scheme (Ungerboeck, 1982; Hatamian et al., 1998; IEEE, 2015) as a systematic way to generate multiple encodings of input features.

As shown in Figure 2, the channel encoder flattens the input features by them into $2 \times 2$ patches of features, then, starting from the upper left feature and ending at the lower left feature, ordering them in a sequence going in the clockwise direction. The features are traversed twice in order to avoid the initialization length of the channel code. This particular scheme is used because it focuses on local relations between features. Exploration of other flattening schemes is left for future research.

The features in the CIFAR-10 dataset are represented by eight bits. The flattened features are fed to the convolutional encoder, which produces one extra bit out of the two least significant bits of the eight bits representing each feature. The 4-D 5-PAM TCM symbol mapper then maps each nine bits into four equidistant 5-PAM symbols, which are then mapped to 12 bits by the bit mapper. The bit mapper uses different symbol-to-bit mappings to generate different encodings of the input features, and the matrix used for generating these encodings is given in Appendix C.7. Each encoding has the same size as the original input samples. Figure 5 in Appendix C.1 shows three CIFAR-10 images and four of their encodings, which are arbitrarily chosen. As seen in this figure, each encoding conveys a different view of the input features, which helps the source code (learned classification function) model relations between the features that are useful for the image classification task. Note that using channel codes on the input features is not a data-augmentation technique: the encodings are appended to the input features, not treated as new input samples. These encodings enable the classifier to learn from the set of available input samples a source code that is sufficiently complex for its classification task. As in a data-transmission or data-storage system, the source code is designed for the most efficient representation of the data, which is the set of available input features for the classification task at hand, and the channel code is independently designed for the channel. This combination is key to achieving generalization in deep learning, and how best to design a channel code for a given classification task is an intriguing future research direction.

## 3 EXPERIMENTAL RESULTS

Let the set of available input samples subjected to common corruptions and adversarial perturbations belong to the empirical sample space from which input samples are drawn during inference. To show that using channel codes on the input features results in learning a more general source code with respect to the true source code, we conduct experiments on the CIFAR-10 and CIFAR-10-C (Hendrycks & Dietterich, 2019) datasets to show increased robustness to common corruptions and adversarial perturbations. For CIFAR-10 and CIFAR-10-C, we train uncoded VGG-11 and VGG-16 models, encoded VGG-11 and VGG-16 models, and an uncoded ResNet-18 model. The VGG networks are modified only by adding the encoder and increasing the number of input channels. The encoded models use the same training criterion as the uncoded models, namely the cross-entropy loss. The training setup and the achieved test accuracies are given in Appendix C.2.

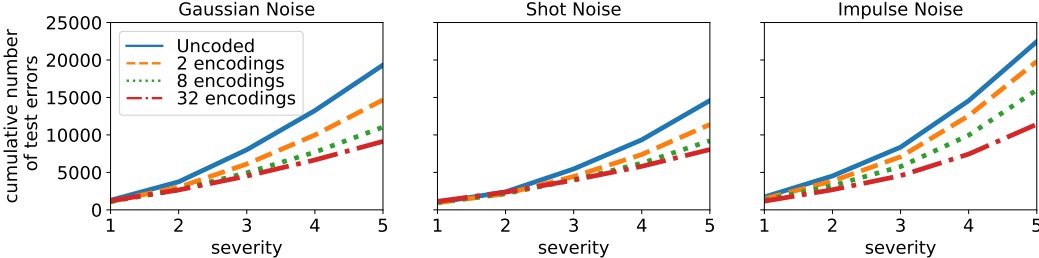

Figure 3: The uncoded VGG-11 model and encoded VGG-11 models tested on the corrupted samples in the CIFAR-10-C dataset.

In all experiments conducted on the encoded models, we use arbitrary encodings. The input samples are corrupted or perturbed before they are input to the encoded models, as the uncorrupted or unperturbed input samples would not be accessible by a neural network in a real-world application. Increasing the number of encodings may reduce the generalization error, but at the expense of increased run time. However, encoding the training and test samples is a one-time process that can be done prior to training, unlike adversarial training, which requires generating perturbed input samples in each epoch. In Appendix C.6, we show that increasing the number of input channels does not, as such, confer robustness to Gaussian noise or to PGD. Designing efficient input codes for a given classification task considering the generalization error and the required number of encodings is a direction for future research. To the best of the authors' knowledge, there is no other published method that can achieve robustness to both common corruptions and adversarial perturbations.

## 3.1 ROBUSTNESS TO COMMON CORRUPTIONS

The set of available input samples may be subjected to common corruptions before reaching a real-world image classifier. For example, Gaussian noise can appear in low-lighting conditions, and shot noise is caused by the discrete nature of light. To show robustness to such corruptions, we conduct experiments on the CIFAR-10-C and CIFAR-10 datasets. We use four common corruptions in our experiments, namely Gaussian noise, shot noise, impulse noise, and speckle noise.

The CIFAR-10-C dataset consists of the 10,000-sample CIFAR-10 test set subjected to five different noise levels, called severity, so it has 50,000 samples in all. As shown in Figure 3, increasing the number of arbitrary encodings concatenated to the original input features increases robustness to Gaussian noise, shot noise, and impulse noise. The results for speckle noise is given in Appendix C.3. For example, when test samples are subjected to impulse noise with a severity level of 4, we see a sharper increase in the number of test errors for the uncoded VGG-11 model than that for the VGG-11 model with 32 encodings. Note that the vertical axis in these plots is cumulative: the number of test errors made at the previous severity level is added to that at the current severity level. Table 1 in Appendix C.4 compares the encoded VGG-11 model with 32 encodings with previously published methods on the CIFAR-10-C dataset, which shows that the encoded VGG-11 model achieves the highest inference accuracy (defined in Appendix B) against shot noise with a severity level of 5 compared with all the other works listed in this table. Additional experimental results on Gaussian noise are shown in Figure 6 in Appendix C.3.

## 3.2 ROBUSTNESS TO ADVERSARIAL PERTURBATIONS

To show robustness to adversarial perturbations without adversarial training, we conduct experiments on the CIFAR-10 dataset. We use the white-box PGD attack (Madry et al., 2018) and transfer attacks from an uncoded VGG-16 and an uncoded ResNet-18 model to evaluate the adversarial robustness of the encoded VGG-16 models. The results for the black-box boundary attack (Brendel et al., 2018) are given in Appendix C.3. The white-box PGD attacks use the gradient of the loss function with respect to the *uncoded* input features in the encoded VGG-16 models because the channel encoder is part of the encoded VGG-16 models; i.e., the only input to the encoded model is the uncoded input features. The encoder is simply a new layer of the neural network architecture, whose outputs are computed directly from the uncoded input features. Changing the outputs of the encoder layer is

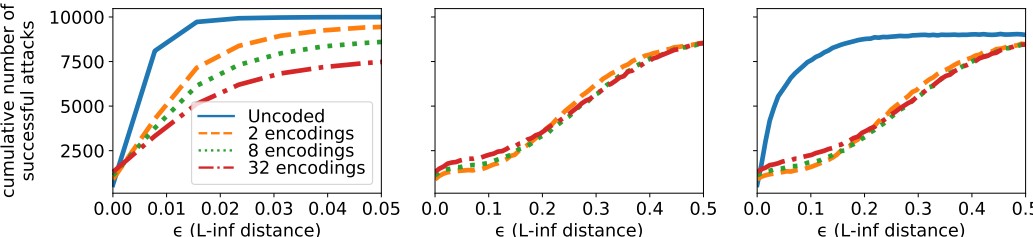

Figure 4: Robustness is tested with samples perturbed by the white-box PGD attack (left), transfer PGD attack generated on the uncoded VGG-16 model (middle), transfer PGD attack generated on ResNet-18 model (right).

tantamount to changing the outputs of any other layer of the model, which is a threat model that falls out of the scope of our work.

For the CIFAR-10 experiments, we use different numbers of encodings, and robustness to all adversarial perturbations in our experiments systemically increased with an increasing number of arbitrary encodings concatenated to the input features. Figure 4 shows the results for the white-box PGD and transfer PGD attacks. The plot on the left shows the increase in robustness to white-box PGD starting from a random perturbation around the natural example and using 20 iterations and a step size of $0.003$. For example, at an $\ell_\infty$ distance of $0.031$, the inference accuracy of the VGG-16 model with 32 encodings is $30.19\%$ while keeping a test accuracy of $87.38\%$. To test the robustness of the encoded VGG-16 models against transfer attacks using the same PGD settings, we generate adversarial examples on the uncoded VGG-16 model and uncoded ResNet-18 model. As before, the encoded VGG-16 models show more robustness with an increasing number of encodings as shown in the middle and right plots of Figure 4. For example, when adversarial examples generated on the ResNet-18 model are used to test the robustness of the uncoded VGG-16 model and the encoded VGG-16 models, at an epsilon of $0.2$ ($\ell_\infty$), the inference accuracy of the uncoded VGG-16 model is $12.25\%$, whereas that of the VGG-16 model with 32 encodings is $63.87\%$.

Table 2 in Appendix C.5 compares the encoded VGG-16 model with previously published defenses on the CIFAR-10 dataset (Zhang et al., 2019). The adversarial attack type used for all the works listed in this table is the white-box PGD starting from a random perturbation around the natural example, using 20 iterations, a step size of $0.003$, and an epsilon of $0.031$ ($\ell_\infty$). It can be observed that this work achieves a sizable inference accuracy $A_{\text{inference}}$ of $30.19\%$ while keeping the highest test accuracy $A_{\text{test}}$ of $87.38\%$ among all the works listed in this table, and importantly *does not use adversarial training*. Generating input encodings can be done just once prior to training, whereas adversarial training requires generating adversarial examples in each epoch, which is expensive when an iterative method such as PGD is used (Madry et al., 2018). The difficult problem of achieving adversarial robustness in a real-world application requires a holistic approach. Our approach, which does not depend on adversarial training, can be readily used in combination with adversarial training and other known methods to achieve greater robustness to adversarial perturbations.

## 4 CONCLUSION

We presented a theoretical and experimental framework for defining and understanding generalization in deep learning, defined as the difference between training and inference errors. The theoretical findings and experimental results show that a learned classification function must be sufficiently complex for a classification task in order to be closer to the true classification function. Another insight from this study is that concatenating encodings of input features to the original input features helps to achieve generalization in deep learning by enabling the classifier to learn relations between features not captured by the original inputs. Experiments demonstrate that a model trained on arbitrarily encoded input features is more robust to common corruptions and adversarial perturbations and that using more encodings may be beneficial to minimize the generalization error. Designing input codes to help a DNN learn a more general classification function with a minimum number of encodings is an intriguing research direction to achieve reliability in machine learning.

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

## A  PROOF OF MATHEMATICAL STATEMENTS

**Proof of Lemma 1.**  For classification task $T$, a learning algorithm is asked to produce the true output function $f(\cdot) : \mathbb{X}^n \to \mathbb{A}$. There exists a source code $C$ for a random variable $\vec{X}$, which is also a mapping from the sample space $\mathbb{X}^n$ of $\vec{X}$ to the $m$-ary signal alphabet $\mathbb{A}$ from which a class $u$ is drawn. The true output function $f(\cdot)$ is equivalent to the source code $C$ for the random variable $\vec{X}$ because their domain $\mathbb{X}^n$ and codomain $\mathbb{A}$ are equal and the image of both functions is the same for each input sample $\vec{x}$ in the domain $\mathbb{X}^n$.

**Proof of Corollary 1.**  If the Kolmogorov complexity $K(\vec{x})$ of an input sample $\vec{x}$ is larger than the number of bits required to describe the class $u$ to which it is mapped, which is at most $\lceil \log_2 m \rceil$ bits, then some information about the input sample $\vec{x}$ is lost. Satisfying this condition, the true source code $C$ is a lossy compressor.

**Proof of Theorem 1.**  The normalized information distance

$$D_{\mathrm{I}}(C, \tilde{C}) = \frac{\max(K(C|\tilde{C}), K(\tilde{C}|C))}{\max(K(C), K(\tilde{C}))} \tag{5}$$

is a universal cognitive similarity metric that minorizes all other admissible distances up to a negligible additive error term. This means that decreasing the normalized information distance $D_{\mathrm{I}}(C, \tilde{C})$ ensures that the true source code $C$ and the learned source code $\tilde{C}$ are more similar; i.e., the learned source code $\tilde{C}$ is more general with respect to the true source code $C$. In a real-world setting, because the empirical sample space $\mathbb{X}^n$ may be too large, the learning algorithm sees an input sample $\vec{x}_{\mathrm{S}}$ drawn from a subset $\mathbb{X}_{\mathrm{S}}^n$ of $\mathbb{X}^n$; i.e., $\mathbb{X}_{\mathrm{S}}^n \subset \mathbb{X}^n$. Put differently, the set $\mathbb{X}_{\mathrm{S}}^n$ of available input samples on which a neural network is trained and tested is a subset of the empirical sample set $\mathbb{X}^n$ which the trained neural network sees during inference. This means that true source code $C$ bears information of all possible relations between input features that are useful for the classification task $T$, whereas the learned source code $\tilde{C}$ bears information of a subset of all possible relations between the input features. The Kolmogorov complexity of the true source code is thus larger than that of a source code learned from the set of available input samples by a sufficiently high-capacity neural network, which can memorize its input samples (Zhang et al., 2017); i.e., $K(C) > K(\tilde{C})$. Therefore,

$$\min_{\tilde{C}}(D_{\mathrm{I}}(C, \tilde{C})) = \min_{\tilde{C}} \max(K(C|\tilde{C}), K(\tilde{C}|C)) \tag{6}$$

is an optimization problem for the generalization of the learned source code $\tilde{C}$ with respect to the true source code $C$.

**Proof of Theorem 2.**  Any encoding $E_i : \mathbb{X}_{\mathrm{S}}^n \to \mathbb{Y}_{\mathrm{S}}^n$ that bears information useful for the classification task $T$ that is not entirely represented by the subset $\mathbb{X}_{\mathrm{S}}^n$ of uncoded input samples; i.e., $\mathbb{Y}_{\mathrm{S}}^n \not\subseteq \mathbb{X}_{\mathrm{S}}^n$, when concatenated with the uncoded input samples $\vec{x}_{\mathrm{S}}$, increases the Kolmogorov complexity of the learned source code, which is now called $\tilde{C}_E$, because a sufficiently high-capacity neural network

can memorize its input samples (Zhang et al., 2017). Put differently, the Kolmogorov complexity $K(\tilde{C}_{\mathrm{E}})$ of the source code $\tilde{C}_{\mathrm{E}}$ learned from a concatenation $\{\vec{x}_{\mathrm{S}}, E_i(\vec{x}_{\mathrm{S}})\}$ of uncoded and encoded input samples is larger than that of the source code $\tilde{C}$ learned from uncoded input samples alone if the encodings bear information of relations between input features that are not represented by the uncoded input samples. As the the true source code $C$ bears information of all possible relations between input features, the Kolmogorov complexity $K(\tilde{C}_{\mathrm{E}})$ of the source code $\tilde{C}_{\mathrm{E}}$ learned from a concatenation $\{\vec{x}_{\mathrm{S}}, E_i(\vec{x}_{\mathrm{S}})\}$ of uncoded input samples and their encodings bearing information of a subset of all possible relations between input features is upper bounded by the Kolmogorov complexity $K(C)$ of the true source code $C$; i.e., $K(C) > K(\tilde{C}_{\mathrm{E}})$. In other words, a sufficiently high-capacity neural network can memorize its input samples (Zhang et al., 2017) without being assisted by encodings $E_i$. However, the encodings $E_i$ bear information of relations between input features, which help to increase the Kolmogorov complexity of the learned source code if they are useful for the classification task $T$; i.e., they are contained in the empirical sample set $\mathbb{X}^n$, of the neural network and if the information in the mappings contained in the input code, which is used to generate the encodings $E_i$, is not represented in the set $\mathbb{X}_{\mathrm{S}}^n$ of available input samples. The conditional Kolmogorov complexities $\{K(C|\tilde{C}), K(\tilde{C}|C)\}$ are thus both larger than $\{K(C|\tilde{C}_E), K(\tilde{C}_E|C)\}$, respectively, because the program that computes how to go from $\tilde{C}_E$ to $C$ is shorter in length than the program that computes how to go from $\tilde{C}$ to $C$. The same holds in the reverse direction. Therefore, $\max(K(C|\tilde{C}_E), K(\tilde{C}_E|C)) < \max(K(C|\tilde{C}), K(\tilde{C}|C))$, which results in $D_{\mathrm{I}}(C, \tilde{C}_E) < D_{\mathrm{I}}(C, \tilde{C})$. The source code $\tilde{C}_E$ learned from the concatenation $\{\vec{x}_{\mathrm{S}}, E_i(\vec{x}_{\mathrm{S}})\}$ is thus more general than the source code $\tilde{C}$ learned from $\vec{x}_{\mathrm{S}}$.

**Proof of Proposition 1.** As the normalized information distance $D_{\mathrm{I}}(C, \tilde{C}_E)$ is not effectively computable, it can be approximated for practical purposes by the normalized compression distance

$$D_{\mathrm{C}}(C, \tilde{C}_E) = \frac{Z(\{C, \tilde{C}_E\}) - \min(Z(C), Z(\tilde{C}_E))}{\max(Z(C), Z(\tilde{C}_E))}, \tag{7}$$

where $Z$ is a real-world compressor. The learning algorithm sees an input sample $\vec{x}_{\mathrm{S}}$ drawn from a subset $\mathbb{X}_{\mathrm{S}}^n$ of $\mathbb{X}^n$ as the empirical sample space $\mathbb{X}^n$ may be too large. Because a sufficiently high-capacity neural network can memorize its input samples (Zhang et al., 2017), the compressed size of the true source code is larger than that of the learned source code; i.e., $Z(C) > Z(\tilde{C}_E)$. At perfect training accuracy, the compressed size $Z(\{C, \tilde{C}_E\})$ of the concatenation $\{C, \tilde{C}_E\}$ is equal to $Z(C)$, as $\tilde{C}_E$ is a partial function of $C$. For a sufficiently high training accuracy, we can consider $|Z(\{C, \tilde{C}_E\}) - Z(C)|$ to be negligible for the purposes of generalization. As the generalization condition $D_{\mathrm{I}}(C, \tilde{C}_E) < D_{\mathrm{I}}(C, \tilde{C})$ is not effectively computable, an equivalent effectively computable condition is useful for practical purposes. As $D_{\mathrm{I}}(C, \tilde{C}_E) < D_{\mathrm{I}}(C, \tilde{C}) \iff D_{\mathrm{C}}(C, \tilde{C}_E) < D_{\mathrm{C}}(C, \tilde{C})$ for the purposes of generalization, the effectively computable condition

$$\frac{Z(\{C, \tilde{C}_E\}) - \min(Z(C), Z(\tilde{C}_E))}{\max(Z(C), Z(\tilde{C}_E))} < \frac{Z(\{C, \tilde{C}\}) - \min(Z(C), Z(\tilde{C}))}{\max(Z(C), Z(\tilde{C}))} \tag{8}$$

is equivalent to

$$Z(\tilde{C}_E) > Z(\tilde{C}). \tag{9}$$

**Proof of Proposition 2.** By the Proof of Proposition 1, the effectively computable optimization problem for the generalization of $\tilde{C}_E$ with respect to $C$ is

$$\min_{\tilde{C}_E} D_{\mathrm{C}}(C, \tilde{C}_E) = \max_{\tilde{C}_E} Z(\tilde{C}_E), \quad \forall \tilde{C}_E : Z(\tilde{C}_E) < Z(C). \tag{10}$$

## B  DEFINITIONS

**Inference Accuracy.** The classification accuracy measured on a subset of the empirical sample set $\mathbb{X}^n$, which may be subjected to common corruptions or adversarial perturbations and which may be out of distribution of the training set, is defined as inference accuracy.

The definition of inference accuracy can be contrasted with that of test accuracy by considering that the former is measured on a subset of the empirical sample set $\mathbb{X}^n$ which consists of corrupted or perturbed samples which may be out of distribution of the training set and that the latter is measured on the test set which consists of uncorrupted and unperturbed samples that are presumed to come from the same distribution as the training set.

**Generalization Error.** The difference between the training error measured on the training set and inference error measured on a subset of the empirical sample set $\mathbb{X}^n$, which may be subjected to common corruptions or adversarial perturbations and which may be out of distribution of the training set, is defined as the generalization error.

This definition is different from that of prior works (Neyshabur et al., 2017; Kawaguchi et al., 2017), which define generalization error as the difference between the training error measured on the training set and test error measured on the test set.

**Generalization.** A learned classification function is said to be more general with a decreasing generalization error.

This definition is different from that of prior works (Neyshabur et al., 2017; Kawaguchi et al., 2017), which define a learned classification function to be more general with a decreasing difference between the training error measured on the training set and test error measured on the test set.

**Source Code.** A source code $C$ for a random variable $\vec{X}$ is a mapping from the sample space $\mathbb{X}^n$ of $\vec{X}$ to an $m$-ary signal alphabet $\mathbb{A}$.

Source codes can be designed for the most efficient representation of the data (Cover & Thomas, 1991). Channel codes appropriate for a channel can be designed separately and independently. This combination is as efficient as any other method that can be designed by considering both problems together. We refer the reader to a textbook such as (Cover & Thomas, 1991) for a detailed understanding of source codes.

**Kolmogorov Complexity.** The Kolmogorov complexity $K_U(x)$ of a string $x$ with respect to a universal computer $U$ is defined as

$$K_U(x) = \min_{p:U(p)=x} l(p),\qquad(11)$$

where $p$ denotes a program, and $l(p)$ denotes the length of the program $p$.

Thus, $K_U(x)$ is the shortest description length of $x$ over all descriptions interpreted by computer $U$. We fix such a universal computer $U$ as reference and write $K_U(x) = K(x)$. We refer the reader to a textbook such as (Cover & Thomas, 1991) and (Bennett et al., 1998; Cilibrasi & Vitányi, 2005) for a detailed understanding of Kolmogorov complexity.

## C    Supplementary Experimental Information

### C.1    Encoded CIFAR-10 Images

Figure 5 shows three CIFAR-10 images and four of their encodings, which are arbitrarily chosen. This figure shows that each encoding conveys a different view of the input features, which helps the learned source code model relations between the features that are useful for the image classification task.

### C.2    Training Setup

All models are trained in PyTorch with 16 random initializations. We train the networks over 450 epochs with a batch size of 128 and with a dynamic learning rate equal to 0.1 until epoch 150, 0.01 until epoch 250, and 0.001 until epoch 450 (Kuang, 2019). A test accuracy of 92.54% is achieved for the uncoded VGG-11 model, and 92.12%, 91.45%, and 90.19% for the VGG-11 model with 2, 8, and 32 encodings, respectively. A test accuracy of 94.15% is achieved for the uncoded VGG-16 model,

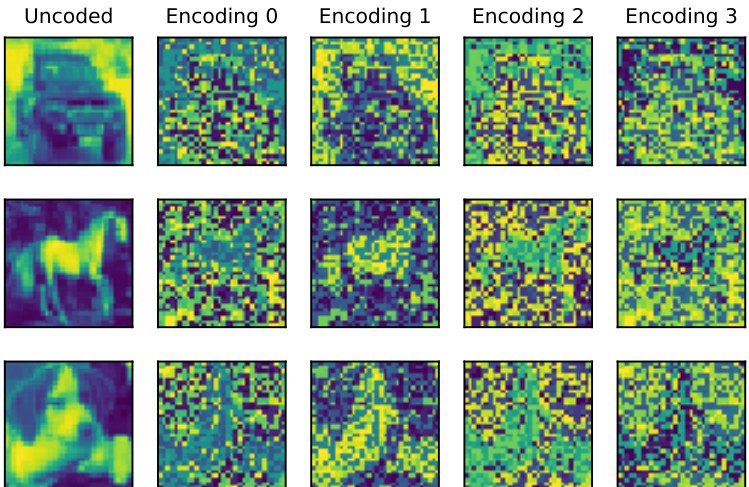

Figure 5: CIFAR-10 uncoded and encoded images. Top, middle, and bottom rows correspond to red, green, blue channels, respectively.

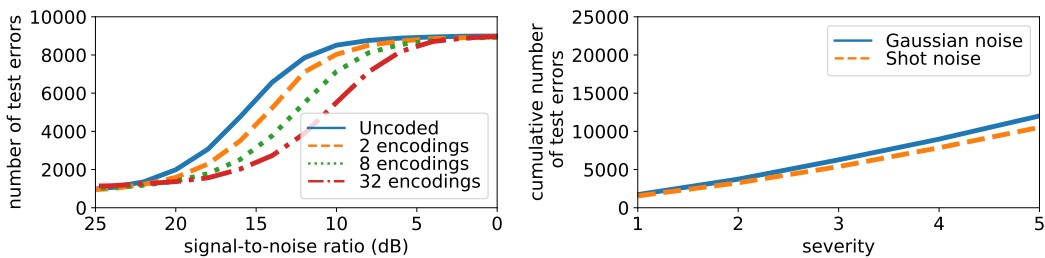

Figure 6: The uncoded VGG-11 model and encoded VGG-11 models tested on the CIFAR-10 test set corrupted by Gaussian noise (left) and the VGG-16 model with 32 encodings tested on the samples in the CIFAR-10-C dataset corrupted by Gaussian noise and shot noise (right).

and 91.11%, 88.93%, and 87.38% for the VGG-16 model with 2, 8, and 32 encodings, respectively. The uncoded ResNet-18 model, which is used for transfer attacks, achieves 95.20% test accuracy.

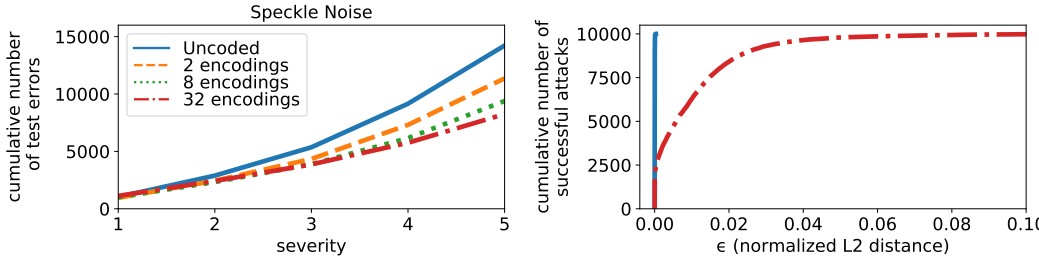

Figure 7: The uncoded VGG-11 model and encoded VGG-11 models tested on the samples in the CIFAR-10-C dataset corrupted by speckle noise (left) and the uncoded VGG-16 model and encoded VGG-16 model tested on the samples in the CIFAR-10 dataset perturbed by the black-box boundary attack (right).

## C.3 ADDITIONAL EXPERIMENTS

To show the robustness of the encoded VGG-11 models to Gaussian noise beyond the noise levels included in the CIFAR-10-C dataset, we apply Gaussian noise with zero mean and variance $\sigma_w^2$ to the CIFAR-10 test set. The average input-feature energy equals

$$\frac{1}{kn} \sum_{i=0}^{kn-1} \vec{x}_i^2, \tag{12}$$

where $\vec{x}_i$ is a feature of the input sample $\vec{x}$, $k$ is the number of input samples in the test set, and $n$ is the number of features in an input sample. We define the signal-to-noise ratio to be

$$\frac{1}{kn\sigma_w^2} \sum_{i=0}^{kn-1} \vec{x}_i^2. \tag{13}$$

In Figure 6, we show on the left plot that increasing the number of arbitrary encodings concatenated to the input features significantly increases robustness to Gaussian noise applied to the CIFAR-10 test set with signal-to-noise ratios from 25 to 0 dB. For example, at a signal-to-noise ratio of 12 dB, the inference accuracy of the VGG-11 model with 32 encodings is 61.15%, whereas that of the uncoded VGG-11 model is 21.49%. On the right plot, the experimental results for the VGG-16 model with 32 encodings tested on the samples in the CIFAR-10-C dataset corrupted by Gaussian noise and shot noise are given. The results indicate that using a larger encoded model does not necessarily confer more robustness to such common corruptions as Gaussian noise and shot noise than a smaller encoded model.

In Figure 7, the results of the experiments conducted on the CIFAR-10-C dataset corrupted by speckle noise and black-box boundary attack experiments on the CIFAR-10 dataset are shown. On the right plot, as with the other type of common-corruption experiments, we see that increasing the number of encodings increases robustness to speckle noise. On the left plot, we see that the encoded model is significantly more robust to the boundary attack than the uncoded model. For example, at a normalized $\ell_2$ distance of 0.01, an inference accuracy of approximately 50% is achieved by the model with 32 encodings, whereas the inference accuracy of the uncoded model already drops to 0% at an $\ell_2$ distance much closer to 0.

## C.4 PERFORMANCE COMPARISON: COMMON CORRUPTIONS

Table 1 compares the encoded VGG-11 model with 32 encodings with previously published methods on the CIFAR-10-C dataset.

At a severity level of 5, the encoded VGG-11 model achieves the highest inference accuracy against shot noise compared with all the other works listed in this table, which use a ResNet-18 or ResNet-26 model. The highest inference accuracy (77.30%) against Gaussian noise is attained by the adversarial logit pairing (ALT) method published in Kannan et al. (2018), but the test accuracy of this method is 83.50%, whereas the encoded VGG-11 model achieves the second highest inference accuracy (75.22%) against Gaussian noise with a test accuracy of 90.19%. Our results seem to indicate that using a larger number of encodings improves robustness to common corruptions, so the inference accuracy achieved by the channel-coding method may be improved by merely increasing the number of encodings or designing higher performance codes.

## C.5 PERFORMANCE COMPARISON: ADVERSARIAL PERTURBATIONS

Table 2 compares the encoded VGG-16 model with previously published defenses on the CIFAR-10 dataset. The experimental results do not imply that encoded input features are more robust to common corruptions and adversarial perturbations than uncoded features. The encoded input features simply bear information of relations between input features that are not represented by the uncoded input features, so a source code learned from a concatenation of uncoded and encoded input features bears more information of the true source code than a source code learned from uncoded input features alone. A source code learned from a concatenation of uncoded and encoded input features is thus more robust than a source code learned from uncoded input features alone.

Table 1: Comparison of the inference accuracy of the encoded model with that of prior methods on the CIFAR-10-C dataset corrupted by Gaussian noise, shot noise, and impulse noise with a severity level of 5.

| Reference | Method | Network | $A_{\text{test}}$ | $A_{\text{inference}}$ Gaussian Noise | $A_{\text{inference}}$ Shot Noise | $A_{\text{inference}}$ Impulse Noise |
|---|---|---|---|---|---|---|
| Birodkar et al. (2019) | RPO | ResNet-18 | **95.70%** | 25.00% | 32.70% | 21.90% |
| Sun et al. (2019) | JT | ResNet-26 | 91.90% | 50.60% | 54.70% | 46.60% |
| Sun et al. (2019) | TTT | ResNet-26 | 92.10% | 54.40% | 58.20% | 50.00% |
| Sun et al. (2019) | TTTO | ResNet-26 | 91.80% | 74.20% | 77.40% | 69.40% |
| Kannan et al. (2018) | ALP | ResNet-26 | 83.50% | **77.30%** | 77.10% | **71.70%** |
| This work | CC | VGG-11 | 90.19% | 75.22% | **77.58%** | 60.18% |

RPO: replacing pooling operator; JT: joint training; TTT: test-time training; TTTO: test-time training online; ALP: adversarial logit pairing; CC: channel coding.

Table 2: Comparison of the encoded model with prior defense models under white-box PGD attacks with 20 iterations, a step size of 0.003, and an epsilon of 0.031 ($\ell_\infty$) on the CIFAR-10 dataset.

| Reference | Defense type | Adversarial training | $A_{\text{test}}$ | $A_{\text{inference}}$ |
|---|---|---|---|---|
| Kurakin et al. (2017a) | regularization | Yes | 85.25% | 45.89% |
| Madry et al. (2018) | robust optimization | Yes | 87.30% | 47.04% |
| Wong et al. (2018) | robust optimization | Yes | 27.07% | 23.54% |
| Zhang et al. (2019) | regularization | Yes | 84.92% | **56.61%** |
| This work | channel coding | **No** | **87.38%** | 30.19% |

## C.6 IMPACT OF INCREASING THE NUMBER OF INPUT CHANNELS OF A DNN ON ITS ROBUSTNESS

To study the impact of increasing the number of input channels of the uncoded VGG-11 and VGG-16 models, we conducted experiments on the encoded VGG-11 and and VGG-16 models that use identical encodings; i.e., the input features are replicated across additional input channels (the "encoders" are just identity functions). In Figure 8, we see on the left that increasing the number of input channels of the uncoded VGG-11 model confers no robustness to Gaussian noise whatsoever. The plot on the right shows that increasing the number of input channels of the uncoded VGG-16 model does not confer robustness to white-box PGD either.

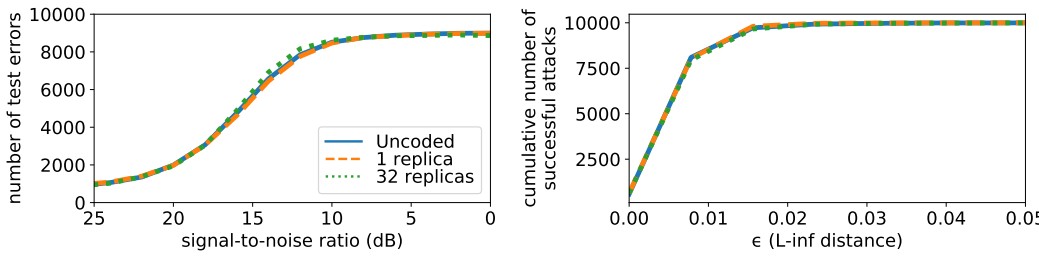

Figure 8: Impact of increasing the number of input channels in the uncoded models on robustness. Robustness to Gaussian noise (left) and the PGD attack (right) is tested by providing identical samples from the CIFAR-10 test set to the increased number of input channels.

## C.7 SYMBOL-TO-BIT MAPPING

The bit mapper in Figure 2 uses the matrix

$$
\begin{bmatrix}
0 & 1 & 0 & 0 & 0 & 1 & 0 & 0 & 0 & 0 & 1 & 1 & 1 & 0 & 0 \\
1 & 1 & 1 & 0 & 1 & 1 & 0 & 0 & 0 & 0 & 1 & 0 & 1 & 1 & 0 \\
1 & 0 & 0 & 0 & 1 & 1 & 0 & 0 & 1 & 1 & 0 & 1 & 0 & 0 & 0 \\
0 & 0 & 0 & 1 & 0 & 0 & 0 & 1 & 0 & 1 & 1 & 0 & 1 & 1 & 1 \\
1 & 0 & 1 & 1 & 0 & 0 & 0 & 0 & 1 & 1 & 1 & 1 & 0 & 1 & 1 \\
1 & 1 & 0 & 1 & 1 & 1 & 0 & 0 & 1 & 0 & 0 & 0 & 1 & 0 & 1 \\
1 & 0 & 1 & 1 & 1 & 0 & 0 & 1 & 0 & 1 & 0 & 0 & 0 & 0 & 1 \\
0 & 1 & 1 & 1 & 0 & 1 & 0 & 1 & 0 & 0 & 0 & 1 & 1 & 1 & 0 \\
0 & 0 & 0 & 0 & 1 & 0 & 0 & 0 & 1 & 0 & 1 & 1 & 1 & 0 & 0 \\
0 & 0 & 0 & 1 & 1 & 1 & 0 & 1 & 1 & 0 & 1 & 0 & 1 & 1 & 0 \\
0 & 0 & 1 & 1 & 0 & 0 & 0 & 1 & 1 & 1 & 0 & 1 & 0 & 0 & 0 \\
0 & 1 & 0 & 0 & 0 & 0 & 1 & 0 & 0 & 1 & 1 & 0 & 1 & 1 & 1 \\
0 & 0 & 1 & 1 & 0 & 1 & 1 & 0 & 0 & 1 & 1 & 1 & 0 & 1 & 1 \\
0 & 0 & 1 & 1 & 1 & 0 & 1 & 1 & 1 & 0 & 0 & 0 & 1 & 0 & 1 \\
0 & 1 & 0 & 1 & 0 & 1 & 1 & 1 & 0 & 1 & 0 & 0 & 0 & 0 & 1 \\
0 & 1 & 0 & 0 & 1 & 1 & 1 & 0 & 1 & 0 & 0 & 1 & 1 & 1 & 0 \\
0 & 1 & 0 & 0 & 0 & 0 & 0 & 0 & 1 & 0 & 1 & 1 & 1 & 0 & 0 \\
1 & 1 & 1 & 0 & 0 & 0 & 0 & 1 & 1 & 0 & 1 & 0 & 1 & 1 & 0 \\
1 & 0 & 0 & 0 & 0 & 1 & 0 & 1 & 1 & 1 & 0 & 1 & 0 & 0 & 0 \\
0 & 0 & 0 & 0 & 1 & 0 & 1 & 0 & 0 & 1 & 1 & 0 & 1 & 1 & 1 \\
1 & 0 & 1 & 0 & 0 & 1 & 1 & 0 & 0 & 1 & 1 & 1 & 0 & 1 & 1 \\
1 & 1 & 0 & 0 & 0 & 1 & 1 & 1 & 1 & 0 & 0 & 0 & 1 & 0 & 1 \\
1 & 0 & 1 & 0 & 1 & 0 & 1 & 1 & 0 & 1 & 0 & 0 & 0 & 0 & 1 \\
0 & 1 & 1 & 0 & 1 & 0 & 1 & 0 & 1 & 0 & 0 & 1 & 1 & 1 & 0 \\
0 & 1 & 0 & 0 & 0 & 1 & 0 & 1 & 1 & 0 & 0 & 0 & 1 & 0 & 0 \\
1 & 1 & 1 & 0 & 1 & 1 & 0 & 1 & 0 & 0 & 0 & 0 & 1 & 1 & 0 \\
1 & 0 & 0 & 0 & 1 & 1 & 1 & 0 & 1 & 0 & 0 & 1 & 0 & 0 & 0 \\
0 & 0 & 0 & 1 & 0 & 0 & 1 & 1 & 0 & 0 & 1 & 0 & 1 & 1 & 1 \\
1 & 0 & 1 & 1 & 0 & 0 & 1 & 1 & 1 & 0 & 0 & 1 & 0 & 1 & 1 \\
1 & 1 & 0 & 1 & 1 & 1 & 0 & 0 & 0 & 0 & 0 & 1 & 1 & 0 & 1 \\
1 & 0 & 1 & 1 & 1 & 0 & 1 & 0 & 0 & 0 & 1 & 0 & 0 & 0 & 1 \\
0 & 1 & 1 & 1 & 0 & 1 & 0 & 0 & 1 & 0 & 1 & 0 & 1 & 1 & 0 \\
\end{bmatrix}
$$

to map four 5-PAM symbols into 12 bits. In this symbol-to-bit mapping matrix, the $i^{\text{th}}$ row corresponds to the encoding $E_i$, where $0 \leq i \leq 31$. Each symbol in the 5-PAM symbol alphabet is converted into three bits by using the corresponding three columns in this matrix. For example, the first symbol in the 5-PAM symbol alphabet for the encoding $E_3$ is converted to $[1\ 0\ 0]$ by drawing the bits from the third row and third, fourth, and fifth columns of the symbol-to-bit mapping matrix. After all four of the 5-PAM symbols are converted into their respective three bits, these bits are concatenated to each other, determining the value of the corresponding feature in the encoded sample.

