# OpenReview forum: "A Kolmogorov Complexity Approach to Generalization in Deep Learning"
_ICLR.cc/2020/Conference — Reject_

### Official Review · AnonReviewer3 · 2019-10-23
**Official Blind Review #3**

**Rating:** 1

**Review:**

The paper proposes an approach to generalization for deep networks based on kolmogorov complexity. The normalized information distance and its approximation via a compression algorithm were developed in earlier work, as noted by the authors. So the main contribution seems to be framing the deep learning classifier as a source code and developing a method to minimize the proposed information distance to improve generalization.

I found a few gaps that I think the authors should clarify for me. Why is framing the deep learning classifier as a source code important? It seems to me that the K(C) is the same as K(f) where the f is the function mapping X -> y, whether it is learned or not.

- Moreover, unless I'm missing something, the source code is a way to encode the values of a random variable such that communication is minimized. If the C=f  is a map from X_i -> y, X_i is an image, and y is a scalar, the source code as defined is not encoding X_i, it is simply encoding a part of X_i that is relevant to the classification task.

1. The claim " Because a sufficiently high-capacity neural network can memorize its input samples the Kolmogorov complexity of the true source code is larger than that of the learned source code: i.e., K(C) > K(C~)." needs proof in itself. This stands opposed to the intuition that an over-fit model has larger kolmogorov complexity because it is not "simple".

- Further, how does this square with the claim that the K(C~) is increased by adding encodings? If K(C~) is increased, then I think one needs to show that K(C~)< K(C) even with the encodings added?
- Finally, by adding the encodings, the task has been fundamentally changed to the classification on the dataset [x, E1(x), E2(x) .. ] . So, the insight about minimizing information distance computed between C and C~ applies when the learned and the true "source code" correspond to the new task. This does not seem to be addressed in the theory.

2. For the noise robustness experiments, there are no other baselines for robustness provided.

3. For the adversarial robustness claims, I don't believe that the justification provided in the paper is the necessarily only one. The adversarial attacks are only done on the uncoded image. So, if 'encodings' are robust to these attacks, the network could also be robust. Unless this hypothesis is rejected, the provided theoretical justification and experiments do not exactly match-up.

(writing comments): I felt that the paper could use a better structure with terms like source code defined in a consolidated section.

**Experience Assessment:**

I have read many papers in this area.

**Review Assessment: Checking Correctness Of Derivations And Theory:**

I assessed the sensibility of the derivations and theory.

**Review Assessment: Checking Correctness Of Experiments:**

I assessed the sensibility of the experiments.

**Review Assessment: Thoroughness In Paper Reading:**

I read the paper at least twice and used my best judgement in assessing the paper.

---

> ### Author Response · Authors · 2019-11-13
> **Response to Official Blind Review #3 (1/3)**
>
> RW3: “I found a few gaps that I think the authors should clarify for me. Why is framing the deep learning classifier as a source code important? It seems to me that the K(C) is the same as K(f) where the f is the function mapping X -$>$ y, whether it is learned or not.”
>
> We thank Reviewer 3 for their valuable comments which have improved the clarity and the content of our manuscript. It is true that $K(C) = K(f)$, whether $f$ or $C$ is learned or not. The reformulation of a learning algorithm as a procedure for searching for a source code is to exploit theoretical results from algorithmic information theory and coding theory for deep learning, thereby avoiding the necessity to reinvent theory that is already established in these fields. Given that source codes are designed for the most efficient representation of data (Cover & Thomas, 1991), we exploit the duality of a source code and a channel code to learn a classification function that represents the input features more efficiently for the classification task $T$; i.e., a more general classification function. Showing that a deep learning classifier is a non-uniquely decodable source code is also fundamental to understanding that the normalized information distance between the input features and the output cannot be used to derive a condition for generalization in deep learning. This results from the fact that deriving such a condition would require finding the conditional Kolmogorov complexity $K(\vec{x}|y)$ of the input features with respect to the output, which is impossible because the source code is not uniquely decodable; i.e., the program to go from the output to the input features cannot be found. A necessary and sufficient condition for generalization based on the normalized information distance can hence be found only between a learned source code and the true source code. Using these results from coding theory and algorithmic information theory, we show that the Kolmogorov complexity of the classification function learned from a concatenation of uncoded and encoded input features is greater than that of the classification function learned from uncoded input features alone if the encodings bear information about relations between input features that are not represented by the uncoded input features. A classification function learned from a concatenation of uncoded and encoded input features is thus more general with respect to the true classification function than a classification function learned from uncoded input features alone. We have updated the manuscript to clarify this point.
>
> RW3: “Moreover, unless I'm missing something, the source code is a way to encode the values of a random variable such that communication is minimized. If the C=f  is a map from X\_i -$>$ y, X\_i is an image, and y is a scalar, the source code as defined is not encoding X\_i, it is simply encoding a part of X\_i that is relevant to the classification task.”
>
> Source codes are designed for the most efficient representation of data (Cover & Thomas, 1991). Whether it is designed for a data-transmission or a data-storage system, a source code, whether lossless or lossy, should retain information about the data necessary to accomplish a given task. The same consideration applies to a learning system. The information in the input features of a learning system is represented by the classification function that it learns; thus, a neural network can be viewed as a source code that encodes inputs features for its classification task. We have updated the manuscript to include more information about source codes and a citation to a textbook to help the reader to better understand our theoretical results.

---

> ### Author Response · Authors · 2019-11-13
> **Response to Official Blind Review #3 (2/3)**
>
> RW3: “1. The claim " Because a sufficiently high-capacity neural network can memorize its input samples the Kolmogorov complexity of the true source code is larger than that of the learned source code: i.e., K(C) $>$ K(C$\sim$)." needs proof in itself. This stands opposed to the intuition that an over-fit model has larger kolmogorov complexity because it is not "simple".”
>
> Please see our response to Reviewer 1 for this statement. Whether a model is over-fit or under-fit does not have a consequence for this statement as this statement compares the Kolmogorov complexity of the learned source code with that of the true source code. The true source code bears information of all possible relations between input features useful for its classification task and successfully classifies input samples from the empirical sample set. Whether a model is over-fit or under-fit is conventionally determined on a cross-validation set and/or test set that are/is identically distributed with the training set, all of which are subsets of the empirical sample set. Being more general on such a cross-validation set and/or test set does not as such guarantee generalization on the empirical sample set because the latter may contain corrupted or perturbed samples and/or there may be samples in the empirical sample set that are out of distribution of the cross-validation set and test set. Our manuscript targets learning a classification function that is more general on the empirical sample set, not only on a cross-validation set and/or test set. We have updated the manuscript to clarify this point.
>
> RW3: “Further, how does this square with the claim that the K(C$\sim$) is increased by adding encodings? If K(C$\sim$) is increased, then I think one needs to show that K(C$\sim$) $<$ K(C) even with the encodings added?”
>
> The Kolmogorov complexity $K(\tilde{C}_\mathrm{E})$ of the source code $\tilde{C}_\mathrm{E}$ learned from a concatenation $\{\vec{x}_\mathrm{S},E_i(\vec{x}_\mathrm{S})\}$ of uncoded and encoded input samples is larger than that of the source code $\tilde{C}$ learned from uncoded input samples alone if the encodings bear information of relations between input features that are not represented by the uncoded input samples. As the the true source code $C$ bears information of all possible relations between input features, the Kolmogorov complexity $K(\tilde{C}_\mathrm{E})$ of the source code $\tilde{C}_\mathrm{E}$ learned from a concatenation $\{\vec{x}_\mathrm{S},E_i(\vec{x}_\mathrm{S})\}$ of uncoded input samples and their encodings bearing information of a subset of all possible relations between input features is upper bounded by the Kolmogorov complexity $K(C)$ of the true source code $C$; i.e., $K(C) > K(\tilde{C}_\mathrm{E})$. We have updated the manuscript to clarify this point.
>
> RW3: “Finally, by adding the encodings, the task has been fundamentally changed to the classification on the dataset [x, E1(x), E2(x) .. ] . So, the insight about minimizing information distance computed between C and C$\sim$ applies when the learned and the true "source code" correspond to the new task. This does not seem to be addressed in the theory.”
>
> The definition of a classification task is given in (Goodfellow et al., 2016, section 5.1.1, page 98), which is cited in the revised version of our manuscript. The classification task does not change because it is defined by the mapping between the input features and the output, which remains the same because the only input to the encoded model is the uncoded input features. The encoder is simply a new layer in the neural network architecture such that an encoded model is designed from an encoder and an uncoded model. We have updated the manuscript to clarify this point.
>
> RW3: “2. For the noise robustness experiments, there are no other baselines for robustness provided.”
>
> We have updated the manuscript with the results of prior works. We would like to point out that to the best of our knowledge there is no other published method that can achieve robustness to both common corruptions and adversarial perturbations.

---

> ### Author Response · Authors · 2019-11-13
> **Response to Official Blind Review #3 (3/3)**
>
> RW3: “3. For the adversarial robustness claims, I don't believe that the justification provided in the paper is the necessarily only one. The adversarial attacks are only done on the uncoded image. So, if 'encodings' are robust to these attacks, the network could also be robust. Unless this hypothesis is rejected, the provided theoretical justification and experiments do not exactly match-up.”
>
> The only input to the encoded model is the uncoded input features. The encoder is simply a new layer of the neural network architecture, whose outputs are computed directly from the uncoded input features. Changing the outputs of the encoder layer is tantamount to changing the outputs of any other layer of the neural network architecture, which is a threat model that falls out of the scope of our work. We have not claimed that the encoded input features are more robust to common corruptions and adversarial perturbations. The encoded input features simply bear information of relations between input features that are not represented by the uncoded input features, so a source code learned from a concatenation of uncoded and encoded input features bears more information of the true source code than a source code learned from uncoded input features alone. A source code learned from a concatenation of uncoded and encoded input features is thus more robust than a source code learned from uncoded input features alone. We have updated the manuscript to clarify this point.
>
> RW3: “(writing comments): I felt that the paper could use a better structure with terms like source code defined in a consolidated section.”
>
> We have included the definition of a source code in the Appendix.

---

### Official Review · AnonReviewer2 · 2019-10-23
**Official Blind Review #2**

**Rating:** 8

**Review:**

This paper provides a very interesting viewpoint for understanding the generalization in deep learning, where the concept of generalization is defined as “the difference between training error and inference error”, and covers the concept of adversarial robustness.  More specifically,
1. The author treats the deep model as a source code, and provides some theoretic analysis based on the Kolmogorov complexity. The insight drawn from this theory is: finding a better source code is equivalent to minimize a Kolmogorov complexity term given the true source code; as a result, a learned classification function needs to be sufficiently complex to minimize the generalization gap.
2. The author tries to approximate the normalized information distance and form an effectively computable optimization problem, which suggests that one can use channel codes on input features as additional inputs.
3. The experiments show that using additional encodings improve robustness in several settings, including in the sense of adversarial robustness.


Pros:
1. The writing is pretty clear.
2. The complexity viewpoint provides more insights for understanding generalization, which is something not covered by learning theory.
3. The empirical method that the authors proposed is simple, and computationally friendly.
4. I personally like the way adversarial robustness is covered in this paper, especially when comparing with tons of other adversarial robustness papers.


Cons:
1. Can you provide exact reference for the definition of normalized information distance? It is not directly mentioned in (Bennett 1988) through a quick scan. The normalization seems important for the theoretical analysis, so it would be better to explain where this definition comes from more clearly.


**Experience Assessment:**

I have read many papers in this area.

**Review Assessment: Checking Correctness Of Derivations And Theory:**

I assessed the sensibility of the derivations and theory.

**Review Assessment: Checking Correctness Of Experiments:**

I carefully checked the experiments.

**Review Assessment: Thoroughness In Paper Reading:**

I read the paper at least twice and used my best judgement in assessing the paper.

---

> ### Author Response · Authors · 2019-11-13
> **Response to Official Blind Review #2**
>
> RW2: “Can you provide exact reference for the definition of normalized information distance? It is not directly mentioned in (Bennett 1988) through a quick scan. The normalization seems important for the theoretical analysis, so it would be better to explain where this definition comes from more clearly.”
>
> We thank Reviewer 2 for their positive comments and improving the clarity of our manuscript. The normalized version of the information distance is referred to in Definition 4.1 and Theorem 4.2 in (Bennett et al., 1998) and explicitly given in Equation (IV.1) in (Cilibrasi & Vitanyi, 2005). We have updated the manuscript to clarify this point.

---

### Official Review · AnonReviewer1 · 2019-10-29
**Official Blind Review #1**

**Rating:** 3

**Review:**

The paper proposes a theoretical framework for studying the generalization of deep neural networks in a broader sense; i.e.,  generalizing with underrepresented or unrepresented training/test samples. The main idea is to cast the problem of improving generalization as the problem of minimizing the normalized information distance between the learned source code and the true source code, defined using the Kolmogorov complexity. Built upon this framework, the method of using extended encodings of an input sample is presented, which empirically seems to lead to better generalization in the settings of adversarial attack and corruptions.

While the idea and the results seem to be appealing, and the concept is novel to the mainstream deep learning community, I suspect that some of the proofs in the paper might not be well grounded. For example, in Equation (3), the authors claim that "a necessary and sufficient condition ensuring that ... the learned source code C_0 is more general than the learned source code C1 is ...". However, this is not proved and does not seem obvious to me. Also, in the proof of Theorem 1 in the Appendix, the authors state "Because a sufficiently high-capacity neural network can memorize its input samples (Zhang et al 2017), the Kolmogorov complexity of the true source code is larger than that of the source code". This, to me, is more like some intuition or conjecture, rather than rigorous mathematical proof. As a result, I am skeptical about the theoretical claims made in the paper.


**Experience Assessment:**

I do not know much about this area.

**Review Assessment: Checking Correctness Of Derivations And Theory:**

I assessed the sensibility of the derivations and theory.

**Review Assessment: Checking Correctness Of Experiments:**

I assessed the sensibility of the experiments.

**Review Assessment: Thoroughness In Paper Reading:**

I read the paper at least twice and used my best judgement in assessing the paper.

---

> ### Author Response · Authors · 2019-11-13
> **Response to Official Blind Review #1**
>
> RW:1 “While the idea and the results seem to be appealing, and the concept is novel to the mainstream deep learning community, I suspect that some of the proofs in the paper might not be well grounded. For example, in Equation (3), the authors claim that "a necessary and sufficient condition ensuring that ... the learned source code C\_0 is more general than the learned source code C1 is ...". However, this is not proved and does not seem obvious to me.”
>
> We thank Reviewer 1 for their valuable comments which have improved the clarity and the content of our manuscript. Equation (3) is a direct result of using the normalized information distance as a universal cognitive similarity metric to determine whether learned source code $\tilde{C}_0$ or $\tilde{C}_1$ is more general with respect to the true source code $C$. The normalized information distance (Bennett et al., 1998) is a metric that uncovers all effective similarities between the true source code and a learned source code. Learning a source code that is closer to the true source code under this metric thus ensures achieving generalization. We have updated the manuscript to clarify this point.
>
> RW1: “Also, in the proof of Theorem 1 in the Appendix, the authors state "Because a sufficiently high-capacity neural network can memorize its input samples (Zhang et al 2017), the Kolmogorov complexity of the true source code is larger than that of the source code". This, to me, is more like some intuition or conjecture, rather than rigorous mathematical proof. As a result, I am skeptical about the theoretical claims made in the paper.”
>
> This statement in the Proof of Theorem 1 is equivalent to simply stating that the set $\mathbb{X}_\mathrm{S}^n$ of available input samples on which a neural network is trained and tested is a subset of the empirical sample set $\mathbb{X}^n$ which the trained neural network sees during inference; i.e., $\mathbb{X}_\mathrm{S}^n \subset \mathbb{X}^n$. This means that true source code $C$ bears information of all possible relations between input features that are useful for the classification task $T$, whereas the learned source code $\tilde{C}$ bears information of a subset of all possible relations between the input features. The Kolmogorov complexity of the true source code is thus larger than that of a source code learned from the set of available input samples by a sufficiently high-capacity neural network, which can memorize its input samples (Zhang et al., 2017). We have updated the Proof of Theorem 1 to clarify this point.

---

### Official Review · AnonReviewer4 · 2019-11-03
**Official Blind Review #4**

**Rating:** 3

**Review:**

This paper proposes a novel approach to training classifiers inspired by concepts from algorithmic information theory. Specifically, the inputs are augmented with additional features formed by encoding the image (using a source code). In addition, the paper argues to use the normalized compression distance as a loss criterion when training; this is a computationally tractable proxy for the normalized information distance, which the paper argues is what one really should use to generalize well, but which is not computationally tractable. (In particular, evaluating the Kolmogorov complexity, which is required to compute the normalized information distance, is not tractable.) The idea is demonstrated via experiments training common image classifiers (VGG-11 and ResNet-18) on clean and corrupted versions of the CIFAR-10 dataset, and comparing how they fare against adversarially trained methods on noisy and adversarial examples.

I believe that this paper contains interesting and novel ideas, but there are also some inconsistencies and some points that are vague. I would consider raising my score if the authors address or clarify these issues, which are detailed below.

The introduction discusses generalization very broadly, suggesting that the usual definition of generalization (e.g., accuracy on a hold-out set, drawn from the same distribution as the training set) is not the version of generalization that one should focus on. However, the precise definition of generalization is never provided. The intro also mentions briefly about adversarial examples, and this is the focus of the experiments. It would help to clarify whether the paper focuses on being robust to adversarial examples as a form of generalization, or more precisely, what how is the concept of generalization defined/measured in this paper. (E.g., the first sentence of Sec 2.1 mentions "our running definition of generalization," but no precise definition has yet been provided.

To make the paper self-contained, it would help to recall the formal definition of Kolmogorov complexity; this could be done in the appendix.

Sec 2 begins to make the connection between a classifier and a source code. However, these definitions are not precise. For example, in the usual PAC learning setting one has a distribution D over the training data (x,y). After specifying a particular loss function, the Bayes-optimal classifier is well defined. Does the source code C correspond this classifier? I would guess not, since the paper seems to be introducing a specific loss criterion, but it would be good to clarify this in the paper. Since you are assuming there is a "true output function" f, does this correspond to what is usually referred to as the "realizable" setting (i.e., where y = f(x) is a deterministic function of x)?

Notation is sometimes used without having been defined. For example, what is $\vec{x}_{S}$?

It isn't clear how one can train using the criterion (4), or equivalently, solve the problem set forth in Prop 2, in practice, since the true source code C is not known. Please elaborate on this.

In the experiments, how precisely is the VGG network modified to handle the encoded inputs? Are these just passed as additional input channels? Is the network architecture need to be modified in any other way to account for these additional inputs? Also, what loss criterion is used for training? Cross-entropy? Or something related to normalized information distance? This isn't clear from the discussion in Sec 3.

In Sec 3.2, is it really fair for the PGD attacks to only use the gradient of the loss wrt the uncoded input? Is this still really "white box"? How does the proposed approach perform if the attack also has access to the encoded input? It would be good to report both settings in the paper.

The introduction made connections to other forms of generalization, including domain generalization and domain adaptation. Based on this, I had expected to experiments illustrating the utility of the proposed for these problems.

The existing experiments do make clear that the encoded inputs are more robust to perturbations and aversarial attacks than uncoded inputs. Is the encoding assumed to be performed on the perturbed input, or on an unperturbed input (i.e., before it is perturbed)?

I don't find the results in Table 1 fully convincing that the proposed approach should be preferred over previous approaches. For instance, the approach of Madry et al. (2018) achieves slightly lower test accuracy, but substantially higher inference accuracy. In general, one expects to see a tradeoff here. If one were to plot these points, would the proposed method clearly fall above the Pareto frontier? Minor: Please clarify the definition of "inference accuracy" in the paper.



**Experience Assessment:**

I have read many papers in this area.

**Review Assessment: Checking Correctness Of Derivations And Theory:**

I carefully checked the derivations and theory.

**Review Assessment: Checking Correctness Of Experiments:**

I assessed the sensibility of the experiments.

**Review Assessment: Thoroughness In Paper Reading:**

I read the paper at least twice and used my best judgement in assessing the paper.

---

> ### Author Response · Authors · 2019-11-13
> **Response to Official Blind Review #4 (1/3)**
>
> RW4: “The introduction discusses generalization very broadly, suggesting that the usual definition of generalization (e.g., accuracy on a hold-out set, drawn from the same distribution as the training set) is not the version of generalization that one should focus on. However, the precise definition of generalization is never provided.”
>
> We thank Reviewer 4 for their valuable comments which have improved the clarity and the content of our manuscript. The only difference between the literature and our work in defining generalization is that we define a learned classification function to be more general with a decreasing difference between the training error and inference error, as opposed to the difference between the training error and test error. The distinction between “test error” and “inference error” is that “test error” is measured on the test set, which consists of uncorrupted and unperturbed samples that are presumed to come from the same distribution as the training set, while “inference error'' is measured on a subset of the empirical sample set, which consists of corrupted or perturbed samples which may be out of distribution of the training set. We have updated the manuscript to clarify this point and provided citations to prior works that study generalization for a comparison between the different definitions.
>
> RW4: “The intro also mentions briefly about adversarial examples, and this is the focus of the experiments. It would help to clarify whether the paper focuses on being robust to adversarial examples as a form of generalization, or more precisely, what how is the concept of generalization defined/measured in this paper. (E.g., the first sentence of Sec 2.1 mentions "our running definition of generalization," but no precise definition has yet been provided.”
>
> The focus of our paper is on learning a classification function that is more general with respect to the true classification function of the classification task $T$. We use robustness to both common corruptions and adversarial robustness to measure how well a learned classification function generalizes on the empirical sample set, which contains corrupted or perturbed samples. We have updated the manuscript to clarify this point.
>
> RW4: “To make the paper self-contained, it would help to recall the formal definition of Kolmogorov complexity; this could be done in the appendix.”
>
> We have included the definition of Kolmogorov complexity in the Appendix.
>
> RW4: “Sec 2 begins to make the connection between a classifier and a source code. However, these definitions are not precise. For example, in the usual PAC learning setting one has a distribution D over the training data (x,y). After specifying a particular loss function, the Bayes-optimal classifier is well defined. Does the source code C correspond this classifier? I would guess not, since the paper seems to be introducing a specific loss criterion, but it would be good to clarify this in the paper. Since you are assuming there is a "true output function" f, does this correspond to what is usually referred to as the "realizable" setting (i.e., where y = f(x) is a deterministic function of x)?”
>
> We have not introduced a specific loss criterion. The encoded models use the same criterion as the uncoded model, namely the cross-entropy loss. The true source code $C$ is equal to the true classification function $f(.)$. We have updated the manuscript to clarify this point and included the definition of a source code in the Appendix.
>
> RW4: “Notation is sometimes used without having been defined. For example, what is $\vec{x}_\mathrm{S}$?”
>
> The symbol $\vec{x}_\mathrm{S}$ was already specified in the first place where it was used in the manuscript as denoting the available input features. We have updated the manuscript to emphasize that the available input features $\vec{x}_\mathrm{S}$ are drawn from the set $\mathbb{X}_\mathrm{S}^n$ of available input features; i.e., $\vec{x}_\mathrm{S} \in \mathbb{X}_\mathrm{S}^n$, which is a subset of the empirical sample set $\mathbb{X}^n$.

---

> ### Author Response · Authors · 2019-11-13
> **Response to Official Blind Review #4 (2/3)**
>
> RW4: “It isn't clear how one can train using the criterion (4), or equivalently, solve the problem set forth in Prop 2, in practice, since the true source code C is not known. Please elaborate on this.”
>
> Equation (4) is not a training criterion, but is the normalized compression distance approximating the normalized information distance given in Equation (2). It is used to derive the theoretical results, particularly the use of input codes to achieve generalization, which are illustrated by experiments. Proposition 2 shows that the compressed size $Z(\tilde{C}_E)$ of the source code $\tilde{C}_E$ must be maximized until it reaches the compressed size $Z(C)$  of the true source code $C$ in order to achieve generalization. This statement results from the fact that the source code $\tilde{C}_E$ is a partial function of $C$ at perfect training accuracy, so the true source code $C$ is not needed to be found to show that $Z(C) > Z(\tilde{C}_E)$. In other words, the source code $\tilde{C}_E$ learned from the concatenation $\{\vec{x}_\mathrm{S},E_i(\vec{x}_\mathrm{S})\}$ of the encoded input samples $E_i(\vec{x}_\mathrm{S})$ and the uncoded input samples $\vec{x}_\mathrm{S}$ can be made more general if the encoded input samples $E_i(\vec{x}_\mathrm{S})$ bear information of relations between input features that are not represented by its input samples. We have updated the manuscript to elaborate on this point.
>
> RW4: “In the experiments, how precisely is the VGG network modified to handle the encoded inputs? Are these just passed as additional input channels? Is the network architecture need to be modified in any other way to account for these additional inputs? Also, what loss criterion is used for training? Cross-entropy? Or something related to normalized information distance? This isn't clear from the discussion in Sec 3.”
>
> The VGG network is modified only by adding the encoder and increasing the number of input channels, and the cross-entropy loss is used for training. The normalized information distance given in Equation (2) is used to derive the theoretical results, particularly the use of input codes to achieve generalization, which are illustrated by experiments. We have updated the manuscript to clarify this point.
>
> RW4: “In Sec 3.2, is it really fair for the PGD attacks to only use the gradient of the loss wrt the uncoded input? Is this still really "white box"? How does the proposed approach perform if the attack also has access to the encoded input? It would be good to report both settings in the paper.”
>
> The only input to the encoded model is the uncoded input features. The encoder is simply a new layer of the neural network architecture, whose outputs are determined directly by the uncoded input features. The adversary is assumed to know the encoded model architecture with all of its design details. The PGD attacks are thus white-box. Changing the outputs of the encoder layer is tantamount to changing the outputs of any other layer of the neural network architecture, which is a threat model that falls out of the scope of our work. We have updated the manuscript to clarify this point.
>
> RW4: “The introduction made connections to other forms of generalization, including domain generalization and domain adaptation. Based on this, I had expected to experiments illustrating the utility of the proposed for these problems.”
>
> We compared our work to domain-generalization and domain-adaptation methods to distinguish our approach from the literature. We have updated the manuscript to clarify this point.
>
> RW4: “The existing experiments do make clear that the encoded inputs are more robust to perturbations and aversarial attacks than uncoded inputs. Is the encoding assumed to be performed on the perturbed input, or on an unperturbed input (i.e., before it is perturbed)?”
>
> We have not claimed that the encoded input features are more robust to common corruptions and adversarial perturbations. The encoded input features simply bear information of relations between input features that are not represented by the uncoded input features, so a source code learned from a concatenation of uncoded and encoded input features bears more information of the true source code than a source code learned from uncoded input features alone. A source code learned from a concatenation of uncoded and encoded input features is thus more robust than a source code learned from uncoded input features alone. The input samples are perturbed before they are input to the encoded model, as the unperturbed input samples would not be accessible by a neural network in a real-world application. We have updated the manuscript to clarify this point.

---

> ### Author Response · Authors · 2019-11-13
> **Response to Official Blind Review #4 (3/3)**
>
> RW4: “I don't find the results in Table 1 fully convincing that the proposed approach should be preferred over previous approaches. For instance, the approach of Madry et al. (2018) achieves slightly lower test accuracy, but substantially higher inference accuracy. In general, one expects to see a tradeoff here. If one were to plot these points, would the proposed method clearly fall above the Pareto frontier?”
>
> We do not argue that our method in isolation should be preferred over another method because there is still a large difference between the test error and inference error achieved by any of the published methods on the CIFAR-10 dataset. The difficult problem of achieving adversarial robustness in a real-world application requires a holistic approach, which is why any method that does not depend on adversarial training and that results in a considerable inference accuracy should be regarded as highly valuable because it can be readily used in combination with known methods to achieve greater robustness to adversarial perturbations. We would like to point out that to the best of our knowledge there is no other published method that can achieve robustness to both common corruptions and adversarial perturbations. In the meantime, we improved on the inference accuracy achieved by the encoded models against adversarial perturbations by using the VGG-16 network.
>
> RW4: “Minor: Please clarify the definition of "inference accuracy" in the paper.”
>
> We have included the definition of inference accuracy in the Appendix.

---

> ### Comment · AnonReviewer4 · 2019-11-13
> **Acknowledging responses**
>
> Thank you for the detailed responses and clarifications. I read them and the revised version of the paper. I appreciate the clarifications.
>
> Regarding the response RW4, the idea that this may be used in combination with other techniques is reasonable, but this has not yet been demonstrated (e.g., in Section 3), so it is difficult to judge the value of this proposition.

---

### Decision · Program_Chairs · 2019-12-19

**Decision:**

Reject

**Comment:**

This is an interesting paper that aims to redefine generalization based on the difference between the training error and the inference error (measured on the empirical sample set), rather than the test error. The authors propose to improve generalization in image classification by augmenting the input with encodings of the image using a source code, and learn this encoding using the compression distance, an approximation of the Kolmogorov complexity. They show that training in this fashion leads to performance that is more robust to corruption and adversarial perturbations that exist in the empirical sample set.

Reviewers agree on the importance of this topic and the novelty of the approach, but there continue to exist sharp disagreement in the ratings. Most have concerns about the formalism and clarity in the presentation. Especially given that the paper is 10 pages, it should be evaluated against a more rigorous standard, which doesn't appear to be met. I encourage the authors to consider a rewrite with a goal towards clarity for a more general ML audience and resubmit for a future conference.